# Drought Resistant Resting Cysts of *Paraphysoderma sedebokerense* Preserves the Species Viability and Its Virulence

**DOI:** 10.3390/plants12183230

**Published:** 2023-09-11

**Authors:** David Alors, Sammy Boussiba, Aliza Zarka

**Affiliations:** 1Microalgal Biotechnology Laboratory, the Jacob Blaustein Institutes for Desert Research, Sede-Boker Campus Ben Gurion University of the Negev, Beersheba 8499000, Israel; sammy@bgu.ac.il; 2Departamento de Biología y Químicas, Facultad de Recursos Naturales, Campus San Juan Pablo II, Universidad Católica de Temuco, Temuco 478 0694, Chile

**Keywords:** *Paraphysoderma sedebokerense*, *Haematococcus lacustris*, resting cyst, drought, stress

## Abstract

The blastocladialean fungus *P. sedebokerense* is a facultative parasite of economically important microalgae and for this reason it has gained a lot of interest. *P. sedebokerense* has a complex life cycle which includes vegetative and resting stages. The resting cysts were assumed to play an essential role in survival by resisting drought, but this ability was never tested and the factors that trigger their formation were not evaluated. This study was aimed to induce resting cyst formation and germination in *P. sedebokerense*. At first, we tested the survival of *P. sedebokerense* liquid cultures and found that infectivity is retained for less than two months when the cultures were stored on the bench at room temperature. We noticed that dry cultures retained the infectivity for a longer time. We, thus, developed a method, which is based on dehydration and rehydration of the biomass, to produce, maintain, and germinate resting cysts of *P. sedebokerense* in both saprophytic and parasitic modes of growth. When the dry cultures were rehydrated and incubated at 30 °C, resting cysts asynchronously germinated after 5 h and the “endosporangium” was protruding outside of the cyst. Our method can be used to preserve *P. sedebokerense* for research purposes with the advantage of no need for expensive equipment.

## 1. Introduction

The fungus *Paraphysoderma sedebokerense* belongs to the Blastocladiomycota division, one of the aquatic true fungi [1,2]. It has gained a lot of interest since it is a highly specific parasite of the economically important microalga *Haematococcus lacustris* [1,3,4], the best natural producer of the high-value pigment astaxanthin [5], well known for its benefits in various clinical applications [6]. It can also parasitize *Chromochloris zofingiensis* [3], a producer of high value carotenoids [7], and *Scenedesmus dimorphus* [8] which can be grown for biofuel production [9] but with a lower preference as compared to *H. lacustris* [4]. It is especially a threat for *H. lacustris* outdoor cultures, because of the high lethality to this microalga (100%) and the worldwide distribution of the parasite: Israel [1], USA [8], Portugal [10], South Korea [11], and China [12]. Little is known about *P. sedebokerense* in the environment but, based on isolation localities, it seems to inhabit transient water bodies which suffer wide fluctuations in abiotic conditions. The aquatic true fungi which inhabit those stressful environments, combine resistant structures with stages of rapid growth, and are being considered as a type of extremophiles [13]. The production of resting cyst by Blastocladiomycota fungi confers also resistance to freezing [14]. Recently this ability to resist cryopreservation has been demonstrated in *P. sedebokernse* [15]. The production of resting cysts is only part of their survival strategy which includes the tolerance of extreme environmental conditions and the ability to change the lifestyle from parasites feeding on host cells to saprotrophs feeding on detritus [16]. These survival strategies are comparable with the ones that exist in of *P. sedebokerense*, i.e., the fast life cycle which can be completed as a saprobe or parasite [1] within 12–16 h, as demonstrated in the parasitic lifestyle [4], and includes a stage of resistant cysts, characterized by thick cell walls that are assumed to resist the dry season [1,8]. In aquatic true fungi the formation of resting cyst is favored by adverse conditions and the germination of resting cysts requires a resting period and favorable conditions [17,18]. However, the particular environmental conditions leading the production and maturation of blastocladialean-resistance structures have not been clearly and thoroughly documented.

The closest relatives to *P. sedebokerense* are species of the genus Physoderma, which are plant parasites and their resting cysts are probably the more studied ones within Bastoclaiomycota [19,20,21,22]. Lange and Olson [23] investigated the germination of the resting cysts of the maize pest *P. maydis*—after incubating them in water—and characterized the stages preceding the opening of the operculum and the protrusion of the resting sporangia (see figures 10–13 [23]).

From our own experience we know that *P. sedebokerense* liquid cultures lose their viability and their ability to infect microalgae after prolonged storage, when frequent reinoculation is not conducted (not published). This phenomenon imposes a challenge to researchers, since the frequent inoculation of the fungus is time consuming and the loss of viable fungus enforce re-isolation and strain validation, which implies an effort consuming time and manpower labor. In this study, we investigated the time that it takes a *P. sedebokerense* culture to become inactive, expecting that old cultures would lose the infectivity but not young cultures. By testing fungal samples stored for different periods it was possible to determine a threshold of time from which the activity was lost, but one sample was an exception retaining the activity for a long time. We attributed this retained infectivity to the formation and germination of resting cysts, since this sample was totally dry and we rehydrated it before using it as an inoculum. We, thus, included in this study the following objectives: (1) To develop a protocol to produce and germinate resting cysts; (2) To test the resistance of resting cysts to abiotic stress, such as dehydration, being infective after germination; (3) Testing if resting cysts can be used for storage of cultures and strains of *P. sedebokerense*.

## 2. Results

### 2.1. Evaluation of Infectivity of Samples Maintained at Room Temperature

We evaluated the infectivity of liquid of *P. sedebokerense* monocultures (Figure 1A) and of *P. sedebokerense*-*H. lacustris* cocultures (Figure 1B), maintained at room temperature for six different periods of time, from 1 to 6 months. The infectivity of all collected cultures was tested via inoculation with fresh *H. lacustris* cultures in 12-well plates (Figure 1C,D). We found that after one month on the bench at RT pure and cocultures retained their ability to infect fresh *H. lacustris* cultures. However, after two months of storage, all the cultures lose their infectivity and only one culture showed infectivity (5-month-old, Figure 1(D2)). This culture was occasionally dried during the storage period and it was necessary to add some distilled water to recover the Blastoclad dried biomass from the bottom of the flask before using it for inoculation (Figure 1(B2)).

### 2.2. Observation of Resting Cysts in Stationary Cultures

The formation of resting cysts in pure *P. sedebokerense* culture is not synchronous; in stationary cultures we observed the different stages of the vegetative life cycle including propagules, vegetative cysts, and small proportion of resting cysts. Resting cysts were observed in both pure Blastoclad stationary cultures and coculture with microalgae. In addition to the thicker cell wall, resting cysts can be distinguished by the formation of a big central vacuole (Figure 2, arrows).

### 2.3. Production of Resting Cysts and Germination by Dehydration and Hydration

To induce the formation of resting cysts, we incubated 3 mL logarithmic *P. sedebokerense* cultures (as monoculture or co-culture with *H. lacustris*) in a shaker (140 rpm) incubator at 30 °C for 7–10 days, in CELLSTAR^®^ suspension 6-well plates (Greiner Bio-One, Frickenhausen, Germany), which were unsealed to allow slow dehydration.

Once all the wells in the plate were totally dried, the 6-well plates were removed from the shaker and kept on the bench at RT. Based on the work of Lange and Olson [23] we tried to germinate the resting cysts of the dried plates by the addition of 3 mL sterile distilled water in each well and immediately incubated the plates in the 30 °C shaker (140 rpm) for further sampling and testing. Each culture was examined under the microscope to visualize resting cysts formation and was tested for infectivity.

With our newly designed method, we increased the production of resting cysts (as compared to stationary culture) and we observed their germination. We observed protrusion of the resting cysts content in preparations, as early as 5 h after hydration and up to 20 h. One hour after hydration the resting cysts remain intact (Figure 3A) and only after 5 h we observe the breakage of the outer thick cell wall as a pore, which is progressively opening (Figure 3B–D) and protruding from the inner content as a whole (Figure 3D–G), and finally the aperture forms an obtuse angle (Figure 3H).

The germination of the resting cysts was not synchronous, and in the same preparation we saw germinating resting cysts with their content partially protruded and completely closed resting cysts (Figure 4A). Operculum or lid was not observed, not even in the initial stages of the germination (Figure 3 and Figure 4). The germination process is accompanied by a change in Nile Red fluorescence; the typical yellow stain of the neutral storage lipids disappears, and instead red fluorescence—characteristic to polar lipids—is observed (Figure 4B).

### 2.4. Infectivity Depending on Hydration Time and Storage Time on Bench

We induced cyst formation in six types of cultures, by adding 0.5 mL of logarithmic *P. sedebokerense* cultures (3–4 days old) in BGM (wet *P. sedebokerense*) in an empty well (dry *P. sedebokerense*), in green *H. lacustris*, red *H. lacustris*, *C. zofingiensis,* and *S. dimorphus* cultures. We assayed each culture in a total volume of 3 mL (except for the dry *P. sedebokerense* which was 0.5 mL). We compared the infectivity of the dried samples stored for two weeks on the bench, after they were rehydrated for 5 and 20 h. All tested samples retain infectivity and become infectious towards fresh green *H. lacustris* cultures after rehydration. Three days after inoculating *H. lacustris* cultures with the rehydrated *P. sedebokerense* preparations, the collapse of the algae cells was evident macroscopically and microscopically (all infected cultures had host cells carrying *P. sedebokerense* cysts) (Figure 5). However, the shorter hydration time (5 h) was more effective than the longer hydration time (20 h) in samples of dry *P. sedebokerense* (Figure 5C,D) and green *H. lacustris* infection (Figure 5E,F).

We also tested cultures that were kept dried for two months on the bench. These cultures retained infectivity after hydration of the samples for 5 h (Figure 6), but the collapse of *H. lacustris* cultures was observed only after 5 days. Two samples cause an exceptionally fast (24 h) crush of fresh *H. lacustris* cultures; those samples were wet *P. sedebokerense* and red *H. lacustris*. The rehydrated samples were also inoculated in fresh BGM culture media to measure their viability, and only wet *P. sedebokerense* and red *H. lacustris* were able to grow (not shown).

## 3. Discussion

In this study, we investigated liquid stationary *P. sedebokerense* cultures kept on the bench, and found that loss of infectivity occurs with high probability after two months of storage at room temperature. This is a general phenomenon that we noticed before (not published); however, some cultures could retain the infectivity for long time period (as our sample in Figure 1(D2)). We hypothesized that old samples retaining infectivity can be due to the presence of resting cysts. Resting cysts of *P. sedebokerense* were described as early as *P. sedebokerense* was described [1], both in monocultures and co-culture with *H. lacustris*. Little is known about how resting cysts are produced and what are the conditions that trigger germination of resting cysts. We knew that resting cyst are produced asynchronously and, based on the loss of infectivity of cultures, it is feasible that they germinate stochastically when in liquid culture and eventually any resting cyst in the culture would be empty. For this reason, we tried to develop a method to induce resting cysts production and maintain the resting cysts ungerminated and ready to germinate upon request.

In lake-inhabiting aquatic true fungi, germination of resting cysts could be coupled to the same factors involved in algal blooms [16] since aquatic true fungi epidemic follows phytoplankton blooms [24,25]. Assuming that *P. sedebokerense* and its algal hosts inhabits ephemeral water ponds it is supposed that resting cyst would be produced prior to summer desiccation and germinate when the conditions would be favorable again for both *P. sedebokerense* and their microalgal hosts. Based on the behavior of *Physoderma maydis* which germinates after rehydration [23], we developed a method to induce the formation of *P. sedebokerense* resting cysts by dehydration, followed by germination upon request in distilled water. This method to produce and germinate resting cyst can be applied to Blastoclad monoculture or algal coculture maintaining infectivity and viability for longer periods than liquid cultures stored at RT. We observed resting cyst germination in any sample assayed with our method based on dehydration and rehydration. Although resting cyst are present in cultures at the stationary phase of growth, we assumed that most of the cysts in the dried samples were produced while cultures were dried. A more important finding is that the resting cysts after dehydration remain dormant and viable being infective after hydration.

The germination of *P. sedebokerense* resting cysts is comparable to Physoderma species, where the central vacuole melts before the protrusion of the inner content (endosporangium) through the cell wall [23,26], and our observations suggest that the neutral storage lipids are consumed and polar membrane lipids are produced (change of the Nile Red fluorescence color from yellow to red, Figure 3). Morphologically, the resting cysts of Physoderma species have one side flattened where a lid detach (operculum) during germination, allowing the endosporangium to protrude [23,27], while the resting cysts of *P. sedebokerense* are spherical and the protrusion of the endo-sporangium was observed after breakage (Figure 3), after that stage we observed propagules (amoeboid swarmers) indicating the release. The fact that we and others [8] did not observed a lid or an operculum could be due to a different opening mechanism involving deformation of the cell wall after small pore opening, or it could be due to the loss of the lid in the microscopic preparation, as happened in figure 12 of Lange and Olson [23]. Unlike *P. maydis* in which cell wall forms an acute angle at the operculum after germination [23], in *P. sedebokerense* we observed an obtuse angle at the aperture, suggesting plastic deformations of the cell wall after opening.

We found that the hydration time is an important variable within which to recover and use the dried cultures, being better shorter (5 h) than longer (20 h) hydration periods. We did not observed resting cyst germination before 4 h and we interpret that this time is required for the induction of germination after water permeate into resting cysts. The reduction in virulence with hydration time is attributed to the progressive mortality of propagules in distilled water, in accordance with the fast mortality of propagules in poor media without carbon source—such as PSM—as previously described by Asatryan et al. [28].

When we compared the dehydrated *P. sedebokerense* samples kept on the bench for two weeks and two months, we found that in both cases all the samples retained the infectivity. We attribute the retention of infectivity to the resting cyst formation because the liquid cultures lose infectivity after two months but not the cultures dehydrated-rehydrated with our protocol.

We did not observed the type of propagules (flagellated zoospores or amoeboid swarmers) that were released from the resting sporangia, but Blastocladiales were described as the only fungal order known to exhibit an alternation of haploid and diploid generations [18,28]; according to Letcher et al. [8], in *P. sedebokerense* the diploid generation can be produced from resting (sexual) sporangia. Why the only two samples of wet *P. sedebokerense* and red *H. lacustris* retained the infectivity is unclear, but it can be related to the time needed to desiccate the plate. Voigt et al. [16] reviewed the different strategies used by microalgal parasites including a fast-killing host with higher reproductive success and a long-infection development with resting cyst formation. In our method 7–10 days were needed until the plates were fully dry. It seems that the cocultures with green *H. lacustris* or green *C. zofingiensis* crushed the microalgal cultures and finished their cycle fast, probably before suffering the desiccation stress. The dry *P. sedebokerense* sample without added medium suffered fast desiccation stress and had no time to produce resting cysts, or they did not succeed to germinate due to the lack of nutrient after hydration with distilled water. The *P. sdebokerense* monoculture continued their life cycle until the desiccation stress triggered the resting cyst formation as previously suggested [1] and the coculture of red *H. lacustris* could be a combination of slower infection processes which induce higher resting cyst production as reported before [28]. High resting cyst production is also expected in the culture of Scenedesmus due to the low specificity of *P. sedebokerense* to this microalga [4]. In previous works we found a high proportion of resting cysts compared to vegetative cysts in *P. sedebokerense*—*S. dimorphus* coculture, but we did not quantify how many resting cysts were produced; however, it seems that this is not the case when this culture is dried.

Based on our results and observations, we developed a method which is useful to maintain *P. sedebokerense* cultures and to study the resting cyst formation and germination, either in a saprophytic or parasitic lifestyle. We propose to apply the method to a logarithmic *P. sedebokerense* monoculture or co-culture with *H. lacustris* at the red stage. One great advantage of our method is the simplicity, since there is no need for expensive equipment and it can be adapted to be used in the field on sites of isolation, being an alternative to more sophisticated method to preserve *P. sedebokerense* strains [15]. We observed differences in resting cyst production and germination when infecting the red or the green *H. lacustris* in accordance with the differences in the infectivity against green and red *H. lacustris* showed by previously [28]. We interpret that when *P. sedebokerense* infects red *H. lacustris* it suffers stress that facilitates resting cyst formation.

The observed ability of the *P. sedebokerense* resting cysts to survive dryness for months and to germinate when it is in contact with water suggests adaptation to the environment and to the microalgae that it infects. Thanks to the resting cysts *P. sedebokerense* can not only survive in dry environments but also “follow” the ephemeral ponds inhabiting air-dispersed and widely distributed *H. lacustris* [29,30]. Not only the fungal parasite is able to survive dehydration; we also recovered viable *C. zofingiensis* and *S. dimorphus* when rehydrated the dried samples. These findings are in accordance with reports showing that microalgae can survive in dry soil, even for years [31,32] and can develop thick cell walls and metabolites to resist the dryness and UV [33]. Our results further support suggestions that *P. sedebokerense* is also adapted to ephemeral ponds, resisting drought, and potentially being dispersed by wind. This explains the wide distribution of the parasite (found in the three continents of the North).

In conclusion, we report here a very simple protocol to induce *P. sedebokerense* rest cyst formation and germination upon request. The protocol is useful for both strain preservation and cyst investigations. Additionally, our findings highlight some considerations applicable in microalgal production plants: to use closed bioreactors, when possible, to avoid contact with the pest since it is widely distributed and air-dispersed.

## 4. Materials and Methods

### 4.1. Algal Strains and Maintenance

The algal strain *Haematococcus lacustris* Flotow 1844 em. Wille K-0084 was obtained from Scandinavian Culture Collection of Algae and Protozoa (SCCAP) at the University of Copenhagen, Denmark. *Chromochloris zofingiensis* strain SAG 211-14 was obtained from the Culture Collection of Algae at the University of Gottingen (SAG), Germany, and the *Scenedesmus dimorphus* strain UTEX 1237 was obtained from the Culture Collection of Algae at the University of Texas (UTEX), Austin, USA. The “green” microalgal cultures were grown in mBG11 [28] in an incubator shaker (150 rpm) enriched with CO_2_ (200 mL min^−1^) at 25 °C for 1 week, as previously described [4]. Briefly, *H. pluvialis* cultures were weekly diluted to approximate densities of 2 × 10^5^ cells mL^−1^; *C. zofingiensis* and *S. dimorphus* cultures were diluted once a week to approximately 5 × 10^6^ cell mL^−1^. At these cell densities, cultures reach the stationary stage after one week. The nitrogen-starved *H. lacustris* culture, referred herein as “red” *H. lacustris* due to its color after starvation, was obtained after 1 week inoculation in nitrogen-depleted mBG11 medium; for more details see [28].

### 4.2. Blastoclad Strain and Maintenance

We used the Blastoclad *P. sedebokerense* isolate AZ_ISR isolated at 2019 by Aliza Zarka at Midreshet Ben-Gurion, Israel (Latitude: 30°51′05″ N, Longitude: 34°47′00″ E, Elevation above sea level: 480 m = 1574 ft); the identity of the strain was confirmed by sequencing of ITS (GenBank voucher: MW336992). This strain was isolated from *H. lacustris*-infected outdoor culture, maintained at the same location from which the original strain was isolated at 2008; it also displayed the same physiological characteristics, as we previously reported [1]. We maintained *P. sedebokerense* in a pure culture via weekly 1/10 dilution of logarithmic Blastoclad culture (from 0.2 to 3.6 OD) in liquid Blastoclad Growth Media (BGM) at 30 °C, in an incubator shaker (140 rpm), supplemented with 2% CO_2_, under continuous dim white light (15 μmol photons m^−2^ s^−1^) illumination [28]. Cultures were also maintained on solid BGM under the same conditions. After one week of incubation liquid cultures were stored on a bench at RT.

### 4.3. Conditions of Infection

The infections were carried out in fresh mBG11 medium; 7–10 days old algal cultures were 10-fold diluted and mixed with *P. sedebokerense* inoculum proceeding from 3–4 days old pure Blastoclad cultures 200-fold diluted. The cocultures were maintained at 30 °C, in an incubator shaker (140 rpm), supplemented with 2% CO_2_, under continuous dim white light (15 μmol photons m^−2^ s^−1^). After one week of incubation cocultures were stored on bench at RT.

### 4.4. Nile red Staining and Microscopic Observations

To visualize lipids in *P. sedebokerense*, we used Nile red as published before [3]. Pellets containing 2 × 10^5^ cells were stained with 100–200 mg Nile red l^−1^ in 2% DMSO, mixed, and washed immediately with 1 mL DDW (10 s; 13,400 g). The pellet was re-suspended in 20–40 mL DDW to obtain a dense sample suitable for microscopic observation and mounting. Samples were observed on a Axioskop1 (Zeiss) microscope employing the light filter allowing maximum exposure to the excitation wavelength (450–490 nm) and a 520 nm cut-off filter. To discriminate between vegetative and resting cysts under the microscope it is better to avoid phase contrast. Under bright field, the darker and thicker cell wall of the resting cysts can be distinguished without staining.

## Figures and Tables

**Figure 1 plants-12-03230-f001:**
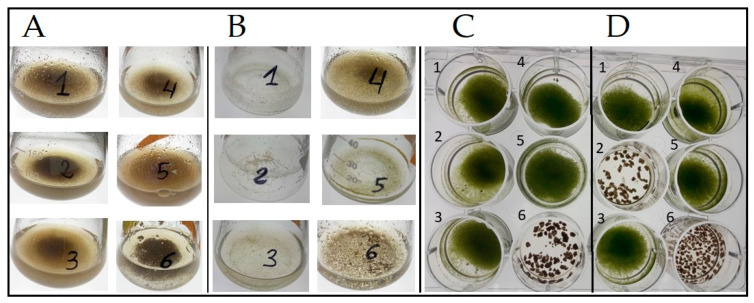
Capability of pure *P. sedebokerense* monocultures (**A**) and *P. sedebokerense*—*H. lacustris* cocultures (**B**), maintained at RT for a period of 1 to 6 months to infect fresh *H. lacustris* cultures (**C** and **D**, respectively). A total of 1–6, 10 mL cultures (**A**,**B**) that were maintained at RT for a period of 6, 5, 4, 3, 2, and 1 month, respectively. These cultures were used to inoculate fresh *H. lacustris* cultures in a 12-well plate, as described in Section 4 Materials and Methods (**C**,**D**). Infection tests were carried out for a period of 7 days. Culture B2 was exceptionally dried during the 5 months storage period and was the only more than one-month-old culture, which caused the collapse of fresh *H. lacustris* culture.

**Figure 2 plants-12-03230-f002:**
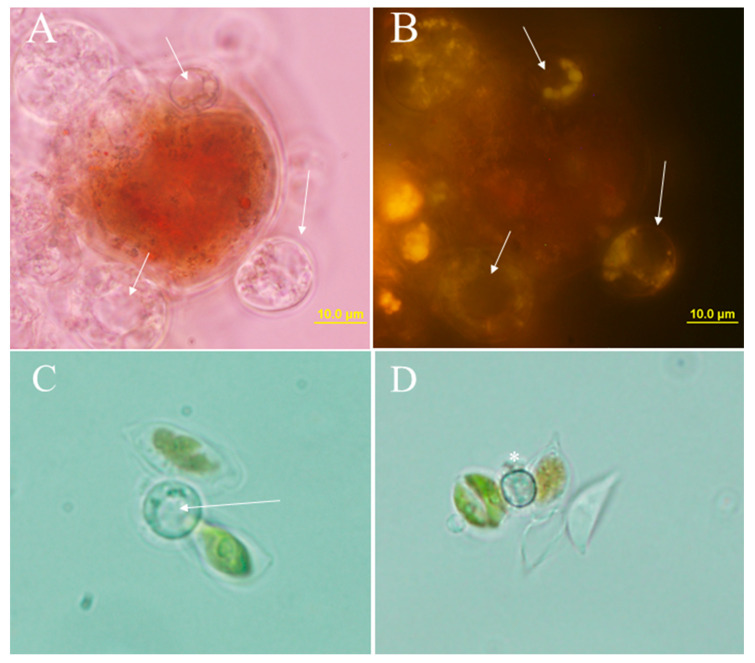
Resting cyst of *P. sedebokerense* formed in microalgal infected cultures. We can see the cyst’s central big vacuole (arrows) in *H. lacustris* coculture under brightfield (**A**) and the cyst’s oil droplets (arrows) stained with Nile Red under fluorescence (**B**). (**C**,**D**), resting cysts in *S. dimorphus* without staining (brightfield), showing the big central vacuole (**C**, arrow) and dark thick cell wall (**D**, asterisk).

**Figure 3 plants-12-03230-f003:**
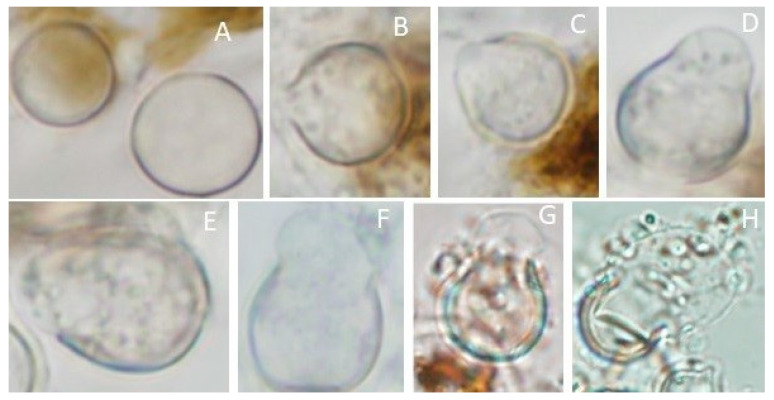
Germination of resting cysts. (**A**) One hour after hydration resting cysts remain closed; (**B**,**C**) Pore with acute angle is opening 5 h after hydration; (**D**–**G**) The pore of the resting cyst is progressively opening, and the content is protruding; (**H**) Finally, the whole “endosporangium” goes out of the resting cyst to liberate the propagules.

**Figure 4 plants-12-03230-f004:**
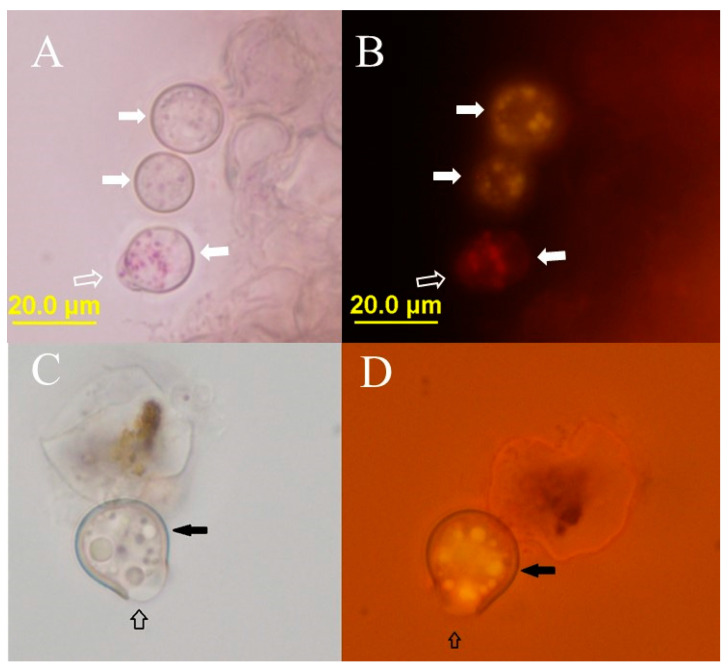
Resting cysts germination in distilled water. Pure *P. sedebokerense* culture (**A**,**B**) and *H. lacustris* co-culture (**C**,**D**) stained with Nile red, under bright field (**A**,**C**) and under fluorescence (**B**,**D**). Dark thick cell wall (solid arrows) and the protrusion of the inner content (hollow arrow) can be observed. In (**B**), the fluorescence of the Nile red is changing from yellow to red (white arrows) in the germinating resting cyst, indicating a change in the composition of fatty acids from neutral to polar.

**Figure 5 plants-12-03230-f005:**
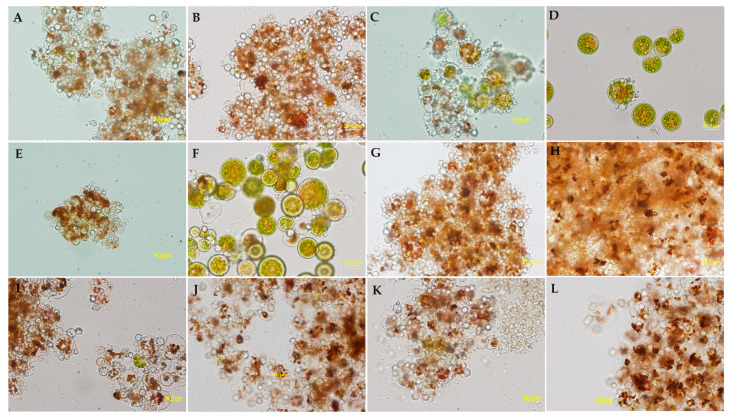
Infectivity of different *P. sedebokerense* cultures (monocultures or Blastoclad-alga cocultures), induced to form resting cysts, towards green *H. lacustris*. The dried Blastoclad cultures were maintained at RT for 2 weeks before testing their infectivity for 3 days. The inocula used were Blastoclad monoculture in BGM hydrated for 5 (**A**) or 20 h (**B**), Blastoclad monoculture without culture media hydrated for 5 (**C**) or 20 h (**D**), Blastoclad coculture with *H. lacustris* in mBG11 hydrated for 5 (**E**) or 20 h (**F**), Blastoclad coculture with red *H. lacustris* in Nitrogen depleted mBG11 hydrated for 5 (**G**) or 20 h (**H**), Blastoclad coculture with *C. zofingiensis* in mBG11 hydrated for 5 (**I**) or 20 h (**J**), and Blastoclad coculture with *S. dimorphus* in mBG11 hydrated for 5 (**K**) or 20 h (**L**).

**Figure 6 plants-12-03230-f006:**
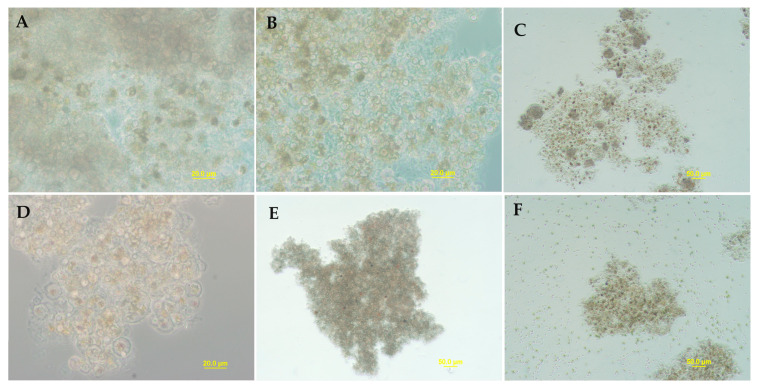
*H. lacustris* cultures crushed 5 days after inoculation with 2 months old dried resting cysts (produced by dehydration protocol) which were germinated via rehydration for 5 h. The inocula used were Blastoclad monoculture in BGM (**A**), Blastoclad coculture with *H. lacustris* in mBG11 (**B**), Blastoclad coculture with *C. zofingiensis* in mBG11 (**C**), Blastoclad monoculture without culture media (**D**), Blastoclad coculture with *H. lacustris* in Nitrogen depleted mBG11 (**E**), and Blastoclad coculture with *S. dimorphus* in mBG11- collapsed *H. lacustris* cells (**F**) can be seen surrounding the clumps while multiple *S. dimorphus* cells from the inoculum are uniformly distributed.

## Data Availability

Data sharing is not applicable to this article.

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
