# Peer review of "Drought Resistant Resting Cysts of Paraphysoderma sedebokerense Preserves the Species Viability and Its Virulence"

_plants, 2023, doi:10.3390/plants12183230_

Round 1
Reviewer 1 Report
Comments for Author,
After I have read your paper carefully, I have found that your paper is interesting and well documented. However, I have some minor comments before accepted this paper.
Specific comments
Please improve abstract section and add more data.
Introduction is needed to detail and please support with properly references.
Actually, some of sentences are needed to rephrase such as ". From our own experience we know that P. sedebokerense liquid cultures lose their activity and their ability to infect microalgae at a certain stage after prolonged storage It is a challenge for the research, since the fungus must be re-isolated, consuming time and manpower labor, and there is some uncertainty about the identity of the re- 58 isolated species or strain. "
Nonetheless, some of methods are needed to extend and please extend briefly of these part.
Results are fine. However, the obtained results is needed to discuss deeply. Otherwise, these parts are superficial. Please focus on during the revision.
Conclusion remark is missed. Please add your future recommandation.
Best Regards
Author Response
Reply to revision has been submitted in word file.

Reviewer 2 Report
This is very interesting research work which added along the previously published work.
Current title.
The ability of Paraphysoderma sedebokerense resting cyst to resist dehydration and their potential to be used as an inexpensive method to preserve strains.
Proposed title.
Drought resistant resting cysts of Paraphysoderma sedebokerense preserves the species, its infectivity and virulence.
· Lines 297-298. Make this more specific (define) for the mentioned name, the location (GPS parameters), and nature of the isolation site(s), point(s) in this statement; “We used the Blastoclad P. sedebokerense isolate AZ_ISR isolated at 2019 by Aliza Zarka at Sede-Boker, Israel”
· It will be very helpful to ascertain a separate outline or a protocol for obtaining resting cysts of this fungus and their usage.
Author Response
Thank you for your kind words and for your valuable suggestions.
We applied modifications suggested by reviewer 1, and also we change the tittle for your proposal and we detailed the isoaliton including GPS coordinates.